# Identification of novel therapeutic inhibitors against E6 and E7 oncogenes of HPV-16 associated with cervical cancer

Saima Younas[1]*, Zaryab Ikram Malik[1], Muhammad Umer Khan[2], Sadia Manzoor[1], Hafiz Muzzammel Rehman[3], Hafiz Muhammad Hammad[3], Shahina Akter[4]*

1 Centre for Applied Molecular Biology (CAMB), University of the Punjab, Lahore, Pakistan, 2 Institute of Molecular Biology and Biotechnology, University of Lahore, Lahore, Pakistan, 3 School of Biochemistry and Biotechnology, University of the Punjab, Lahore, Pakistan, 4 Bangladesh Council of Scientific and Industrial Research (BCSIR), Dhaka, Bangladesh

* saima.camb@pu.edu.pk (SY); shupty2010@gmail.com (SA)

## Abstract

Human Papilloma Virus type 16 (HPV-16) is highly oncogenic with the E6 and E7 oncogenes playing crucial roles in the pathogenesis of HPV-related cervical carcinogenesis. Targeting these oncoproteins with specific inhibitors offers a promising approach for therapeutic intervention. This study aimed to identify potential inhibitors of the HPV-16 E6 and E7 oncoproteins through an in silico approach, providing a foundation for the development of targeted therapies against HPV associated malignancies. We performed virtual screening on a library of 1000 compounds to identify promising candidates. Subsequent molecular docking studies were conducted to assess the binding affinities of the promising candidates. The top-scoring compounds for oncoproteins were then subjected to molecular dynamics simulations to evaluate their stability and interaction profiles. The virtual screening identified 14 promising candidates followed by docking studies. Among these Galangin was identified as a promising inhibitor for the E6 oncogene, while Neoechinulin showed potential as an inhibitor of the E7 oncogene. Our findings suggest Galangin and Neoechinulin with high potential as therapeutic inhibitors of HPV-16 E6 and E7 oncogenes respectively. These inhibitors could contribute significantly to the development of targeted therapies against HPV associated malignancies. However, further in vitro and in vivo investigations are required to use these phytochemicals as antiviral agents against HPV-16.

## Introduction

Cancer has been a leading cause of death in the developing countries throughout the last several decades. According to estimates, the number of deaths from cancer could reach 17.5 million per year by 2050 as a result of exposure to carcinogenic

**Data availability statement:** All relevant data are within the manuscript and its Supporting Information files.

**Funding:** The author(s) received no specific funding for this work.

**Competing interests:** No authors have competing interests.

chemicals, dietary changes, unhealthy lifestyles including smoking, and environmental changes [1]. Among gynecological disorders, cervical cancer is one of the most prevalent worldwide. After lung, colorectal, and breast cancers, it is ranked as the fourth most common cancer in women. However, in developing nations, it is regarded as the second most common type of carcinoma in women. Despite a decline in mortality, cervical cancer is still one of the most common cancers that cause death [2]. Less developed nations accounted for 85% of deaths related to cervical cancer. The elimination of cervical cancer from low-income countries is a significant problem because of a number of variables, including a lack of knowledge, greater exposure to risk factors such as poor hygiene and HPV infection, expensive screening, insufficient immunization campaigns, and sociocultural inhibitions [3].

HPV is major etiological agent of cervical cancer. Healthy cells start to proliferate and take the form of tumors when a cervix cell comes into contact with HPV. Regretfully, the majority of the time, the early stages of this disease do not exhibit any symptoms or indicators. When it worsens, though, menstrual bleeding, vaginal bleeding following intercourse, and copious, red, and sometimes foul-smelling discharge from the vagina are among its symptoms [4].

The human papillomavirus, or HPV, is a tiny, circular virus that carries two strands of DNA. The early gene coding region (E), late gene coding region (L), and long control region are the three separate segments that make up the HPV genome. Based on their ability to cause cancer, HPV strains are categorized as low-risk HPV or high-risk HPV. The International Agency for Research on Cancer (IARC) has classified HPV types 16, 18, 31, 33, 35, 39, 45, 51, 52, 56, 58, 59, and 66 as high-risk human papillomaviruses, and HPV types 6, 11, 40, 42, 43, 44, −53, 54, 61, 72, and 81 as low-risk human papillomaviruses (LR-HPV) [5]. Cervical cancer is mostly caused by high-risk human papillomaviruses (HR-HPV), with the HPV-16 genotype being one of the most common globally. As the primary cause of cervical cancer globally, HPV-16 is regarded as high-risk due to its potent capacity to produce chronic infections and its crucial function in interfering with regular cell cycle regulation. The primary oncogenic factors that determine how quickly the disease progress are the encoding proteins E6 and E7 (oncogenes) from HPV-16. These proteins stimulate proliferation, bring about malignant changes, regulate cell cycle, and make it easier for altered cells to migrate and invade [6].

E6, and E7 are oncogenic, and the other early genes (E1, E2, E3, and E4) encode proteins for regulation and viral replication. The E6 and E7 cancer-causing genes are essential for the onset and advancement of cervical cancer, even though all the encoded genes are vital for replication of virus and life cycle [7].

The structural features of the 151 amino acid HPV-16 E6 protein include two zinc-binding domains with two C-x-x-C motifs apiece, which are essential for the virus's ability to cause cancer12. Additionally, at the protein's carboxyl terminus, its sequence contains a PDZ (PSD95, DLG, and ZO1) class I binding motif (PBM) [6]. E6 is a key player in initiating p53 degradation and inhibitory action. At the E6-p53 complex, E6 will attract ubiquitin, which can then be degraded by the ubiquitin proteasome [7].

Based on anatomical and dynamic features, HPV-16 E7 is highly heterogeneous protein with a length of 98 amino acids. It has been proposed that the translation of E7 is caused by the spliced E6*I transcript. Nevertheless, recent research has shown that circular RNA (circE7), which encompasses the E7 gene, contributes significantly to the levels of E7 protein and its transforming properties, even though it is a less abundant species (~1–3% of the overall E7 transcripts) [8]. A piece of CR1, the whole CR2 region of adenovirus (Ad) E1a, and comparable regions in the SV40 TAg are homologous to N-terminal of E7 of HPV-16 in terms of both sequence and function. The LXCXE where X indicates any amino acid motif links the retinoblastoma tumor suppressor (pRB) and associated pocket proteins to the CR2 homology domain. However, the CR3 domain's residues are also necessary for the best contact. A consensus phosphorylation site for casein kinase II (CKII) at serines 31 and 32 in the case of HPV-16 E7 is located next to this motif [9]. Moreover, it has been demonstrated that E7 forms higher order oligomers, tetramers, and dimers. Moreover, E7 has been demonstrated to be existing in the nucleus, Golgi, and ER subcellular compartments, according to immunofluorescence methods employing antibodies that recognize various HPV-16 E7 epitopes [10]. Moreover, phosphorylation and the proteasome both regulate E7 post-translationally.

Important proteins of the cellular machinery in charge of tumor suppression and cell cycle regulation interact with the E6 and E7 oncoproteins. E6 attaches to TP53, it is ubiquitinated and degraded, which results in a loss of TP53's ability to regulate the cell cycle and a decrease in the expression of proapoptotic proteins like BAX [11]. The transcription factor E2F/DP, which is typically linked to RB1, is released when the E7 protein engages with the RB1 protein. By simulating the physiological conditions of the late G1 phase, this promotes gene transcription, accelerates the cell cycle, and promotes proliferation, all of which lead to oncogenesis [12]. This process involves multiple pathways linked to cancer. Through upstream regulators like RPTK and PI3K, these viral cancer-causing proteins can activate the Akt pathway, which leads to enhanced proliferation. E6 and E7 trigger Wnt pathway, which results in the buildup of β-catenin and may enhance the transcription of genes related to proliferation of cells [11].

Although HPV vaccinations help stop cervical cancer from developing, people who are already infected must be cleared of the virus before receiving a vaccination. Cervarix2 vaccine (HPV16 and 18), Gardasil4 vaccine (HPV 6, 16, and 11), and Gardasil9 vaccine (HPV 6, 16, 11, 31, 18, 45, 33 and 52) are the three preventive vaccines that are currently FDA approved and are used successfully to inhibit ongoing infections and HPV-linked lesions of the cervical cavity. However, the immunization program is not well-established, and most countries aside from a few wealthy ones do not offer the shot [13]. Chemotherapeutic drugs and the application of both ablative and surgical procedures to excise the formed tumors remain the only options for managing cervical cancer and precancerous cervical lesions. Millions of patients, particularly in underdeveloped nations, may not be able to obtain these treatments because to their invasiveness, lack of specificity, and high cost [14]. Consequently, there is an immediate need to create easily available medication therapies that specifically target the oncovirus in order to treat HPV-related illnesses and improve the management and treatment of cervical malignancy and precancerous cervical lesions. Treatments for HPV-associated cancer of the cervical cavity include both targeted and conventional methods. Tumor cells are destroyed by conventional therapies like radiation, chemotherapy, and surgery. In order to treat cancer and address viral processes, advanced therapeutics include vaccines, gene silencing, immunotherapies, and targeted biologics to target E6 and E7.

E6 and E7 can be inhibited by immune-based methods (therapeutic vaccinations), gene silencing (siRNA, CRISPR), antisense technologies, and tiny compounds or peptides that prevent protein interactions [15]. These seek to neutralize oncoproteins that cause cancer in order to restore proper cell cycle regulation and apoptosis. Targeting E6 and E7 directly has a high specificity while it is easier to target downstream pathways with current medications, doing so carries the risk of resistance and off-target consequences. By assuring accurate, consistent, and long-lasting delivery of medications to infected cells, advanced delivery methods improve HPV oncogene targeting. This reduces cytotoxicity and off-target effects while increasing the effectiveness of therapy. By tracking alterations in E6/E7 expression, suppressor genes restoration, cell proliferation/apoptosis indicators, viral clearance, and immune response being activated, the efficacy of HPV

oncogene-targeted treatments is evaluated [16]. Combining direct suppression of HPV oncogenes with immune activation or conventional therapies, combination therapy can increase treatment efficacy, decrease recurrence, and overcome resistance, providing a more thorough and long-lasting approach to controlling HPV-16-driven cervical carcinogenesis.

In this study, 1000 phytochemicals were checked against E6 and E7 oncogenes of HPV-16. Consequently, the goal of the current research is to find potent HPV-16 E6 and E7 inhibitors. In the current study, we used virtual screening to find new, powerful HPV-16 E6 and E7 inhibitors. By anticipating an inhibitor's attachment to target proteins such as HPV-16 E6 and E7, in silico techniques aid in the identification, assessment, and prioritization of possible inhibitors, greatly expediting the drug discovery process prior to in vitro or in vivo confirmation. Three drug-like compounds were identified after docking, resulting in the identification of several chemical entities that provide innovative scaffolds for new families of E6 and E7 HPV-16 inhibitors. The 100 ns MDS investigation made use of top scorer of both E6 and E7 protein. RMSD and RMSF, two metrics that show they share a similar backbone with lead compounds that may function as E6 and E7 HPV-16 inhibitors. Nevertheless, additional in vitro and in vivo studies are required to confirm the in-silico findings. Successful clinical application of E6/E7 inhibitors requires addressing the primary obstacles, which include inadequate structural evidence, biological validation, tumor variations, risk of toxic effects, efficient and targeted delivery, and intricate regulatory pathways.

## Methodology

### Ethical approval

This study was conducted using *in silico* approaches exclusively, without the involvement of human participants, animals, or clinical data. Therefore, no ethical approval or informed consent was required. All computational analyses were performed in accordance with good scientific practice, and the data used were obtained from publicly available databases/resources.

### Retrieval of target proteins

The 3D structure of target protein E6 (PDB ID: 4XR8) of HPV-16 was obtained in PDB format from RCSB PDB database (https://www.rcsb.org/). Sequence of E7 oncogene (NC_001526.4) was retrieved from NCBI (https://www.ncbi.nlm.nih.gov/). Expasy Translate tool (https://web.expasy.org/translate/) was then used to translate the CDS into protein sequence.

### 3D modelling of E7 oncoprotein

I-tasser (https://zhanggroup.org/I-TASSER/) was then used for the 3D modelling of E7 oncogene.

### Structure validation of E7 oncoprotein

The quality of the E7's predicted structure was validated by using Ramachandran plot, Verify3D and ERRAT for its stereo-chemical quality. Structure validation was performed using PROCHECK software.

### Retrieval of ligands

The 3D structure of 1000 different phytochemicals was obtained from PubChem database (https://pubchem.ncbi.nlm.nih.gov/) in the form of sdf files.

### ADME analysis and examination of drug likeness of ligands

The physiochemical, pharmacokinetic characteristics and drug likeness of ligands were determined by web server Swiss ADME (http://www.swissadme.ch/) which has been developed by the Molecular Modeling Group of Swiss Institute of Bioinformatics. Physicochemical descriptors, pharmacokinetic characteristics, drug-like nature, and medicinal chemistry

compatibility are some of the parameters that SwissADME, an open-source server, uses to predict absorption, distribution, metabolism, and excretion qualities of prospective medications. The canonical smiles of ligands obtained from PubChem were submitted to the web server Swiss ADME in order to evaluate their ADME characteristics [17].

### Toxicity prediction of ligands

The hepatotoxicity, carcinogenicity, immunotoxicity, mutagenicity and cytotoxicity of ligands were determined by ProTox-3.0 (https://tox-new.charite.de/protox_II/). ProTox-3.0, which is a web server, used to predict toxicities of small compounds. It integrates molecular resemblance, fragment tendencies, the most prevalent attributes and machine learning algorithms, based on overall 33 algorithms to predict multiple toxicity endpoints including severe toxicity, hepatotoxicity, immunotoxicity, cytotoxicity, mutagenicity, adverse outcomes pathways, carcinogenicity, and toxicity targets [18]. The STopTox (https://stoptox.mml.unc.edu/) was used to determine acute oral toxicity of ligands. STopTox is a quick, dependable, and easy-to-use tool for determining whether a chemical has the potential to trigger acute toxicity. The purpose of the acute toxicity testing is to determine the potential hazards that may arise from brief exposure periods. Several regulatory organizations require the power source of in vivo experiments, also referred to as the "6 pack" assays, to assess a number of acute toxicity elements in humans, such as acute oral toxicity, acute skin toxicity, acute inhalation toxicity, irritation of the skin, corneal irritation and corrosion, and skin sensitivity [19].

### Prediction of binding pocket and molecular docking

CB-Dock 2 was used for the prediction of binding pocket and molecular docking. CB-Dock 2, a protein–ligand blind docking web server. The docking of molecules with the AutoDock Vina (version 1.1.2) was guided by protein-surface-curvature-based cavity recognition approach known as CurPocket [20].

### Molecular dynamics

Bimolecular complex dynamism and conformational changes are investigated in more detail by molecular dynamics simulations. By incorporating the standard Newtonian equation of motion, molecular dynamics simulations often determine how atoms move over time. To investigate the stable state and the conformational steadiness of complexes, we performed 100-ns MD simulations using the Schrodinger Suite's Desmond Simulation tool. The protein-ligand complexes were examined in an orthorhombic rectangular box soaked in solvent and had a minimum gap of 10 Å between the box edges and the protein atoms. To neutralize the models, counter ions were added. NaCl was added at 0.15 M to simulate the physiological state. Using the Nose–Hoover chain thermostat and Martyna–Tobias–Klein barostat approaches, the simulation's isothermal condition (temperature) and isobaric condition (pressure) were configured at 300K and 1atm, correspondingly. Trajectory shots were taken every 20 ps during the 100 ns simulation run. To forecast the ligand's binding orientation, MD trajectories were examined using Desmond's simulation interaction diagram.

## Results and discussion

### 3D modelling of target protein

I-tasser was utilized for modeling the protein sequence. I-tasser predicted the protein models, and the structure with the greatest C-score is the one that was selected. I-TASSER uses the C-score, a confidence score, to gauge the caliber of anticipated models. The framework assembly simulations' convergence parameters and the significance of template alignments are used to calculate it. Typically, the C-score falls between (−5, 2), a greater value symbolizes a model with higher level of confidence and vice versa. 3D model of E7 predicted by I-tasser is shown in Fig 1.

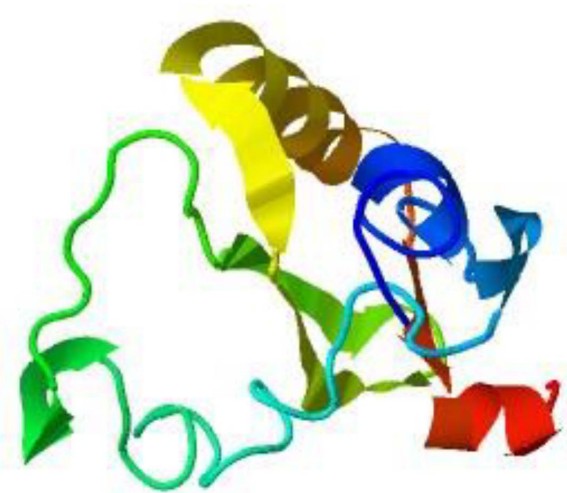

**Fig 1. 3D structure of E7 protein.**

## Structure validation

The Ramachandran plot, VERIFY 3D, and ERRAT tools were used to validate the structure of the modelled E7 protein. Table 1 displays the comparative results of these validations.

### Ramachandran plot

The Ramachandran plot was employed to evaluate the conformational quality of protein backbone torsion angles and study the dihedral angles (Psi and Phi) of amino acids in a protein structure. The majority of the protein's backbone torsion angles appear to be within optimal ranges, as indicated by the fact that more than 90% of amino acid residues in the protein structure are located in this region. This is a reliable sign of an incredibly fine protein structure. Fig 2 shows the RC plot of E7 protein. The plot demonstrates that the most amino acids—represented by black dots—are found in the best-suited area. Table 2 displays the exact percentage of each amino acid found in the various RC plot zones.

### VERIFY 3D

VERIFY3D is an approach that evaluates the overall quality of protein structures by contrasting the protein's amino acid sequence with the three-dimensional model. It shows how well the model fits the experimental data with a profile-window plot. Based on the 3D/1D profile, less than 80% of the amino acids scored >= 0.1, indicating that the E7 protein does not successfully match the necessary amino acid sequences. The VERIFY3D graph of the E7 protein is displayed in Fig 3.

**Table 1. Structure validation of E7 protein.**

| Validation tool | Results |
| --- | --- |
| RC plot | 53.5% |
| ERRAT | 100 |
| VERIFY 3D | 46.94% (Fail) |

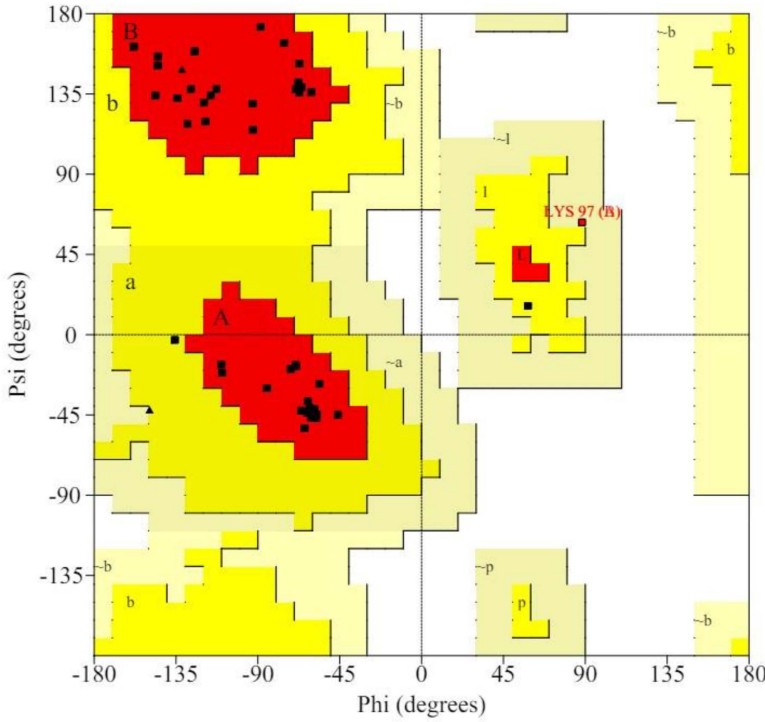

**Fig 2. RC plot of E7 protein.**

**Table 2. Amino acid % in various zones of RC plot.**

| RC plot zones | Amino acids % |
|---|---|
| Most favored zone | 93.2% |
| Additionally allowed zone | 4.5% |
| Generously allowed zone | 2.3% |
| Disallowed zone | 0.0% |

## ERRAT

The ERRAT was designed to evaluate the accuracy of the predicted model. To validate the model, ERRAT employed statistical connections based on atomic interactions between various types of atoms and their non-bonded interactions. Values of 91% and below indicate low-resolution structures, whereas values of 95% and above represent standard high-resolution structures. The E7 protein structure has a very good overall quality, according to the ERRAT score, with most of non-bonded atomic interactions falling within the expected range. Fig 4 displays the expected error graph of the E7 protein, broken down by residue.

## Drug likeness of ligands

A compound that may be used as an oral medication for humans needs to have adequate bioavailability through the oral route. One method of determining the ligand's drug-likeness was to analyze its violation of Rule of five or RO5, as Lipinski had previously documented [21]. Lipinski stated that oral medication could not violate any of following standards including molecular weight or MW < 500 Da, hydrogen bond donor or HBD less than or equal to 5, octanol-water partition coefficient

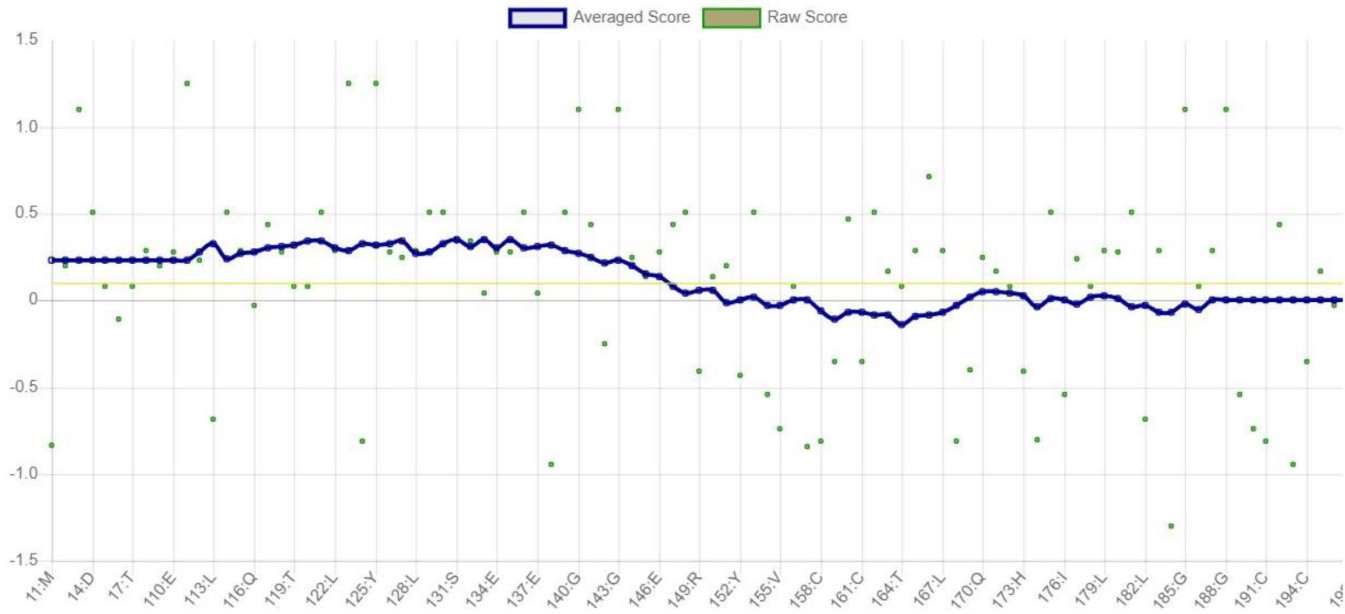

**Fig 3. VERIFY 3D graph of E7 protein.**

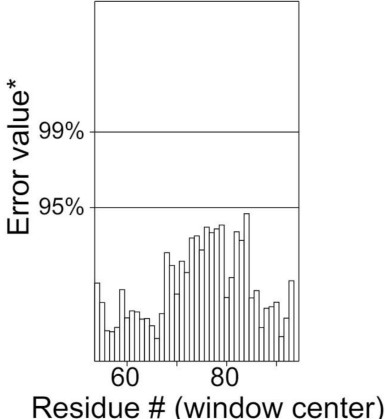

**Fig 4. Error graph of E7 protein.**

or WLogP less than 5, and hydrogen bond acceptor or HBA less than or equal to 10. Only those compounds were selected for further analysis following the Lipinski rule of five. Drug action in the body of humans is significantly influenced by the WLogP parameter of RO5 [22]. All the selected ligands were following Lipinski, Egan, Ghose, Muegge and Veber rules of drug likeness having no violation of these rules.

In a similar vein, calculations of estimated solubility or ESOL utilizing Delaney's approach were verified. Only those compounds were selected which are soluble or have moderate solubility. Compounds having poor solubility were not included in the study. It is advised that drug choices exhibit adequate lipophilicity (WLogP) and outstanding aqueous solubility in water (ESOL) in order to maximize oral absorption and achieve the necessary deposition and effectiveness [23].

The compounds that have been chosen for additional examination can be elicited on target tissues through oral bioavailability. It should be emphasized that although predicting biochemical druggability is a crucial endeavor, it does not provide proof that compounds will function well as medications. It is actually a kind of in silico evaluation of compound's probability of turning into a medication.

Number of rotatable bonds, ease in synthesis, and topological polar surface area or TPSA were among the other important characteristics that were assessed. Ertl's fragmental estimation was used as the basis for calculating TPSA. For optimal absorption, drugs that target tissues other than the CNS should typically have a value of TPSA between 90 and 140 Å2 [24]. Poor adsorption, low permeability of cell membranes, and inadequate absorption when consumed orally are the outcomes of a TPSA greater than 140 Å2 [25]. The amount of rotatable bonds is another factor that affects bioavailability when consumed orally. It has been demonstrated that drugs containing more than ten rotational bonds have lower permeability to membranes and absorption [21]. Consequently, it has been estimated that filtered compounds have good permeability to cell membranes, oral accessibility, and absorption. However, using the computer-aided frequent molecular fragments approach that was developed by Daina and Zoete [26], synthetic accessibility, or SA, of compounds was evaluated in order to assess their ease of synthesis. Their SA calculations indicated that synthesizing them is not difficult.

## ADME characteristics of ligands

In drug research and development, evaluating safety characteristics of drug candidates has grown to be essential to reducing costs and lowering rates of failure related to preclinical phase of the process. Bad pharmacokinetic characteristics have been linked to drug candidates failing in later phases of drug development. In order to identify compounds as chemical leads, it is important to calculate their pharmacological attributes, such as their ADME. This computation also serves as a standard for evaluating manufactured compounds during the process of lead optimization. By focusing on those compounds with a higher likelihood of being successful drugs, weak drug candidates are eliminated through the application of ADMET predictions of the ligands.

The ability of oral medications to be completely absorbed from the digestive system and then distributed to the site of action in the body is one of their key features. Another crucial process is metabolism, which must be properly removed without causing injury in the final stage [27]. ADME findings indicate that selected compounds show high GI absorption in humans, which is consistent with their physicochemical features and highlights their favorable oral absorption. The main barrier dividing the CNS from the bloodstream is the blood-brain barrier, or BBB. The compounds that were chosen were exclusively ones that were anticipated to have low BBB permeability potential. Therefore, when given orally, these compounds probably wouldn't have any neurotoxic effects. To forecast a compound's GI absorption and BBB permeation, we employed the BOILED-Egg approach. This technique has been presented as a reliable forecasting model for determining the degree of polarity and the lipophilicity of tiny compounds [28]. Compounds with high likelihood of being absorbed passively by the gastrointestinal tract are those found in white shadow of the BOILED-Egg visual output. The compounds present in yellow shadow have a high chance of entering the central nervous system by crossing the BBB. Many times, compounds that are not expected to permeate the BBB or be readily absorbed by the gastrointestinal tract fall beyond the graph's range or into its gray area. GI absorption and BBB permeability predictions by Boiled-Egg method of top candidates are shown in Fig 5.

Furthermore, it is anticipated that all selected compounds cannot serve as P-gp substrates. P-glycoprotein is membrane export pump that removes foreign substances, such as medications, from the cell. To prevent interactions between drugs, it is critical in drug development to ascertain early on if potential therapies are P-gp substrates. This process affects the pharmacokinetics of the compounds by causing additional metabolism and elimination. Therapy failure is ultimately

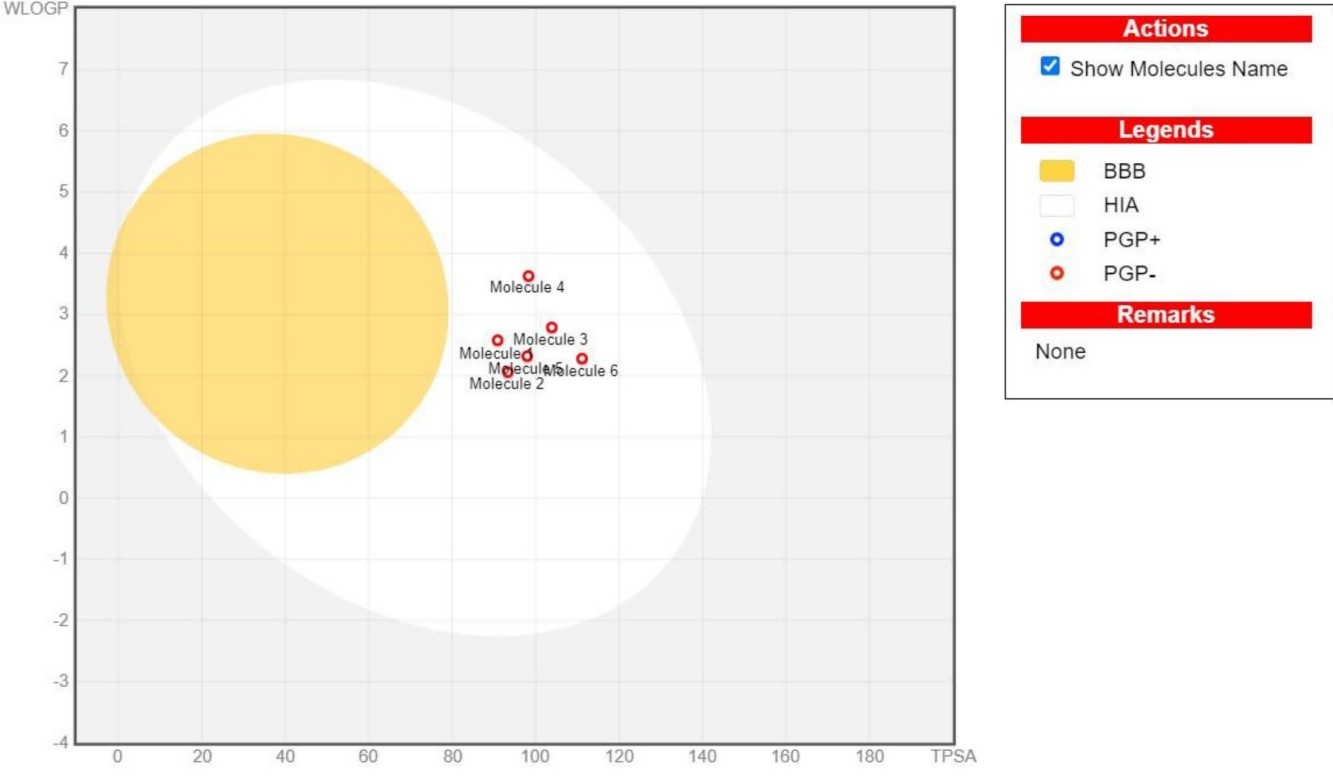

**Fig 5. Estimation of BBB permeability and GI absorption using the BIOLED-Egg technique.**

caused by limited cellular absorption [29]. As a result, our findings imply that selected compounds will probably reach therapeutic concentrations at the places they were intended to.

Enzymes belonging to CYP P450 family (CYP450) are responsible for the metabolism of xenobiotics and are essential for the liver's drug elimination process [30]. Given that small compounds frequently result in interactions between drugs linked to pharmacokinetics, it is imperative to include their CYP450 suppression when assessing toxicity assessments [31]. We evaluated CYP450 suppression profile for each of 5 important isoforms (CYP1A2, CYP2C19, CYP2C9, CYP2D6, and CYP3A4).

When developing new drugs, the pharmaceutical sector takes skin permeability into account in order to assess the risk that may arise from unintentional contact with the skin. Log Kp is used to assess this component, and poorer permeability of the skin is indicated by more negative results [32]. As a result, only compounds with low skin contact risk were chosen.

It needs to be mentioned that each patient's pharmacokinetics are unique and impacted by a range of factors, comprising environment, eating habits, gender, physique, pathophysiology, genetics, and interactions between drugs. In fact, genetic variability in pharmaceutical-metabolizing and carrier genes have been linked to inter-individual heterogeneity in pharmacodynamics as well as pharmacokinetics and changes in the metabolism of drugs [33]. All the selected ligands were following all the above mentioned criteria. ADME characteristics and drug likeness of top three scorers for E6 and E7 oncoproteins are shown in Table 3.

Table 3. ADME analysis & drug likeness along with toxicity prediction of top three candidates for E6 & E7 oncoproteins.

| Target Protein | E6 | | | E7 | | |
|---|---|---|---|---|---|---|
| **PubChem CID** | CID 5281616 | CID 5328779 | CID 6918638 | CID 135540424 | CID 4788 | CID 5280863 |
| **Physiochemical Characteristics** | | | | | | |
| **Formula** | C15H10O5 | C17H14N2O3 | C15H14N2O4S | C15H14N2O4S | C15H14O5 | C15H10O6 |
| **MW** | 270.24 g/mol | 294.30 g/mol | 318.35 g/mol | 391.46 g/mol | 274.27 g/mol | 286.24 g/mol |
| **No. of H-bond donors** | 3 | 5 | 6 | 3 | 4 | 4 |
| **No. of H-bond acceptors** | 5 | 4 | 4 | 4 | 5 | 6 |
| **Rotatable bonds** | 1 | 3 | 3 | 5 | 4 | 1 |
| **TPSA** | 90.9 Å² | 93.35 Å² | 103.88 Å² | 98.31 Å² | 97.99 Å² | 111.13 Å² |
| **Lipophilicity & Solubility** | | | | | | |
| **WLOGP** | 2.58 | 2.06 | 2.79 | 3.63 | 2.32 | 2.28 |
| **Solubility Class** | Soluble | Soluble | Soluble | Soluble | Soluble | Soluble |
| **Pharmacokinetics** | | | | | | |
| **GI absorption** | High | High | High | High | High | High |
| **BBB permeation** | No | No | No | No | No | No |
| **P-gp substrate** | No | No | No | No | No | No |
| **CYP1A2 inhibitor** | Yes | Yes | No | Yes | Yes | Yes |
| **CYP2C19 inhibitor** | No | No | No | Yes | No | No |
| **CYP2C9 inhibitor** | No | Yes | No | No | Yes | No |
| **CYP2D6 inhibitor** | Yes | No | No | Yes | No | Yes |
| **CYP3A4 inhibitor** | Yes | Yes | No | Yes | Yes | Yes |
| **Log Kp** | −6.35 cm/s | −6.38 cm/s | −7.06 cm/s | −5.82 cm/s | −6.11 cm/s | −6.70 cm/s |
| **Drug-likeness** | | | | | | |
| **Lipinski** | 0 violation | 0 | 0 | 0 | 0 | 0 |
| **Ghose** | 0 | 0 | 0 | 0 | 0 | 0 |
| **Egan** | 0 | 0 | 0 | 0 | 0 | 0 |
| **Veber** | 0 | 0 | 0 | 0 | 0 | 0 |
| **Muegge** | 0 | 0 | 0 | 0 | 0 | 0 |
| **Bioavailability Score** | 0.55 | 0.55 | 0.55 | 0.55 | 0.55 | 0.55 |
| **Toxicity Prediction** | | | | | | |
| **Hepatotoxicity** | No | No | No | No | No | No |
| **Immunogenicity** | No | No | No | No | No | No |
| **Cytotoxicity** | No | No | No | No | No | No |
| **Carcinogenicity** | No | No | No | No | No | No |
| **Mutagenicity** | No | No | No | No | No | No |

## Toxicity prediction of ligands

ProTox-II server was used for further filtration of leftover 69 compounds. Any compound displaying hepatotoxicity or carcinogenicity or immunotoxicity or mutagenicity or cytotoxicity was eliminated. After toxicity analysis we were left with 14 compounds. ProTox-II prediction of top three candidates for both E6 and E7 are shown in Table 1. The STopTox was used to predict acute oral toxicity of screened ligands along with their confidence score (Table 4). Predicted fragment contribution of top three candidates for both E6 and E7 in acute oral toxicity are shown in Fig 6.

**Table 4. Acute oral toxicity prediction and confidence scores of top scorers.**

| PubChem CID | Target Protein | Estimation of Oral Toxicity | Confidence Score |
|---|---|---|---|
| CID 5281616 | E6 | Not toxic | 70% |
| CID 5328779 | | Not toxic | 57% |
| CID 6918638 | | Not toxic | 69% |
| CID 135540424 | E7 | Not toxic | 53% |
| CID 4788 | | Not toxic | 78% |
| CID 5280863 | | Not toxic | 70% |

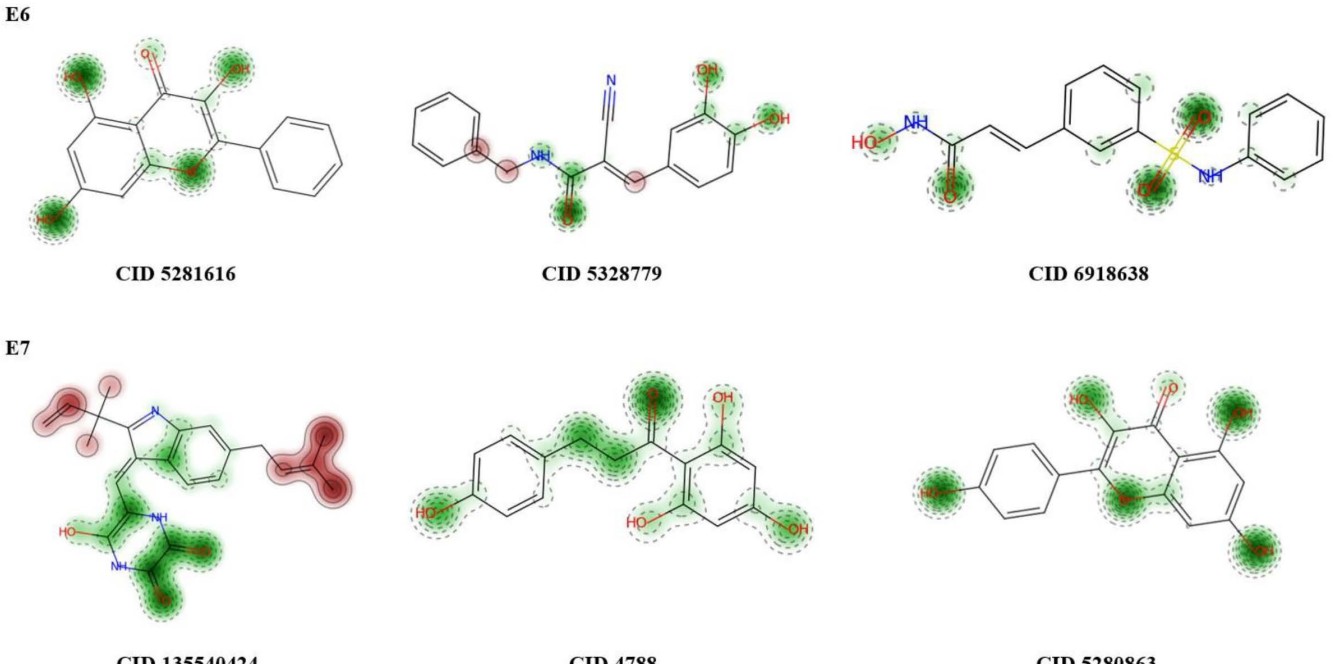

**Fig 6. Contributing domains of top scorers in acute oral toxicity.** Red colored domain contributes toward the oral toxicity.

## Prediction of binding pocket

Binding pockets for target oncoproteins were predicted by using search cavity option of CB-Dock 2. Fig 7 displays the binding pockets of both target proteins.

## Molecular docking

After the prediction of binding pockets target proteins were docked with the 14 ligands filtered after virtual screening. Top three scorers for both oncoproteins are shown in Table 5. CID 5281616 (Galangin) showed highest docking score (−8.8 kcal/mol) with E6 protein followed by CID 5328779 (Tyrphostin B42) with a docking score of −8.3 kcal/mol and CID 6918638 (Belinostat) having a docking score of −8.2 kcal/mol. CID 135540424 (Neoechinulin) showed highest docking score (−8.9 kcal/mol) with E7 protein followed by CID 4788 (Phloretin) with a docking score of −7.8 kcal/mol and CID 5280863 (Kaempferol) having a docking score of −7.2 kcal/mol.

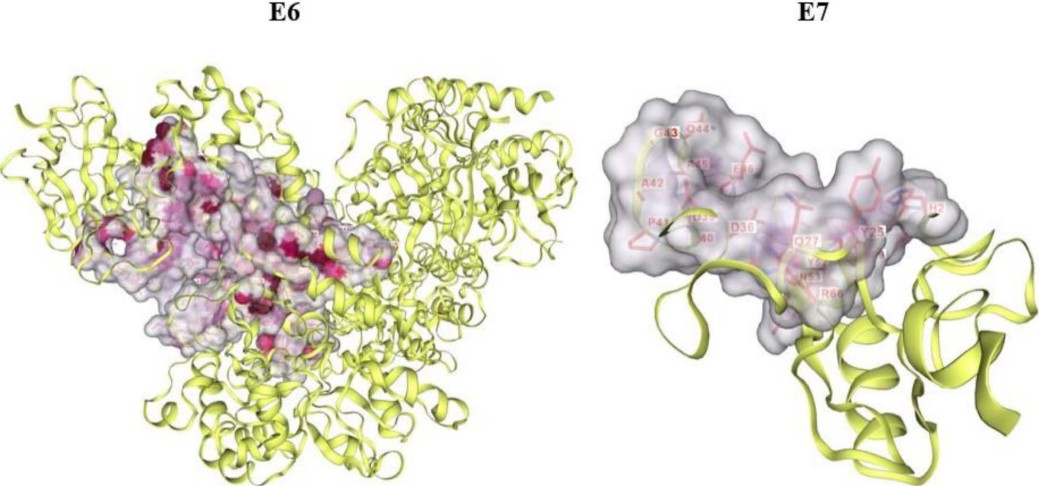

**Fig 7. 3D illustration of binding pockets of E6 and E7.**

**Table 5. Binding affinities and main interactions of top scorers of E6 and E7.**

| PubChem CID | Name of compound | Target Protein | Binding Affinity (S) kcal/mol | Interactions |
|---|---|---|---|---|
| **CID 5281616** | Galangin | E6 | −8.8 | GLU45, GLU46, TRP63,ALA64, ASP66, ARG67, GLU112, GLU154, PRO155, TYR156, PHE157, TRP231, MET331, MET337, TRP341, TYR342, ARG345 |
| **CID 5328779** | Tyrphostin B42 | | −8.3 | LEU379, GLY380, GLU381,GLU382, LEU114, GLN144, TRP146, GLY226, SER227, ASP228, CYS229, THR231,TYR92, LYS94, ASP98, LEU99, LEU100, ARG102, GLY130, ARG131, TRP132 |
| **CID 6918638** | Belinostat | | −8.2 | ASN13, ASP15, LYS16, GLU45, PHE62, TRP63, ALA64, ASP66, ARG67, GLU112, GLU154, PRO155, TYR156, PHE157, VAL262, LEU263, SER338, TRP341, TYR342, ARG345 |
| **CID 135540424** | Neoechi-nulin | E7 | −8.9 | HIS2, TYR25, GLN27, GLU35, ASP36, ASP39, GLY40, ALA42, GLY43, GLN44, ALA45, GLU46, PRO47, HIS51, ASN53, ARG66, CYS68 |
| **CID 4788** | Phloretin | | −7.8 | HIS2, TYR25, GLN27, ASP36, ASP39, GLY40, PRO41, ALA42, GLN44, ALA45, GLU46, HIS51, TYR52, ASN53, ARG66, CYS68, VAL69, GLN70 |
| **CID 5280863** | Kaempferol | | −7.2 | HIS2, GLN27, ASP36, ASP39, GLY40, PRO41, GLN44, GLU46, ALA50, HIS51, TYR52, ASN53, ARG66, CYS68, VAL69, GLN70 |

## Ligand interactions

CID 5281616 (Galangin) formed Pi-cationic interactions with ARG345, GLU 45 and GLU154. It formed Pi-alikyl interactions with PRO155 and Pi-Pi stacking with TYR342 and TYR156. Carbon hydrogen bonding was shown with ARG67 and TRP341. Hydrogen bonding was displayed with ASP66 and TRP63 (Fig 8a). CID 5328779 (Tyr-phostin B42) showed H-bonding with ASP228, SER227, CYS229 and TRP132. It formed Pi-alkyl interactions with LEU100 and ARG131. Carbon hydrogen bonding was shown with GLY130 and Pi-Pi T-shaped interactions were shown with TRP146 (Fig 8b). CID 6918638 (Belinostat) formed H-bonding with GlU122. It formed Pi-Pi stacking with TYR156 and Pi-Pi T-shaped interactions with TYR342. Pi-sulfur interactions were shown with TRP341 (Fig 8c). 2D and 3D interactions of top three candidates (CID 5281616, CID 5328779 and CID 6918638 with E6 are shown in Fig 8.

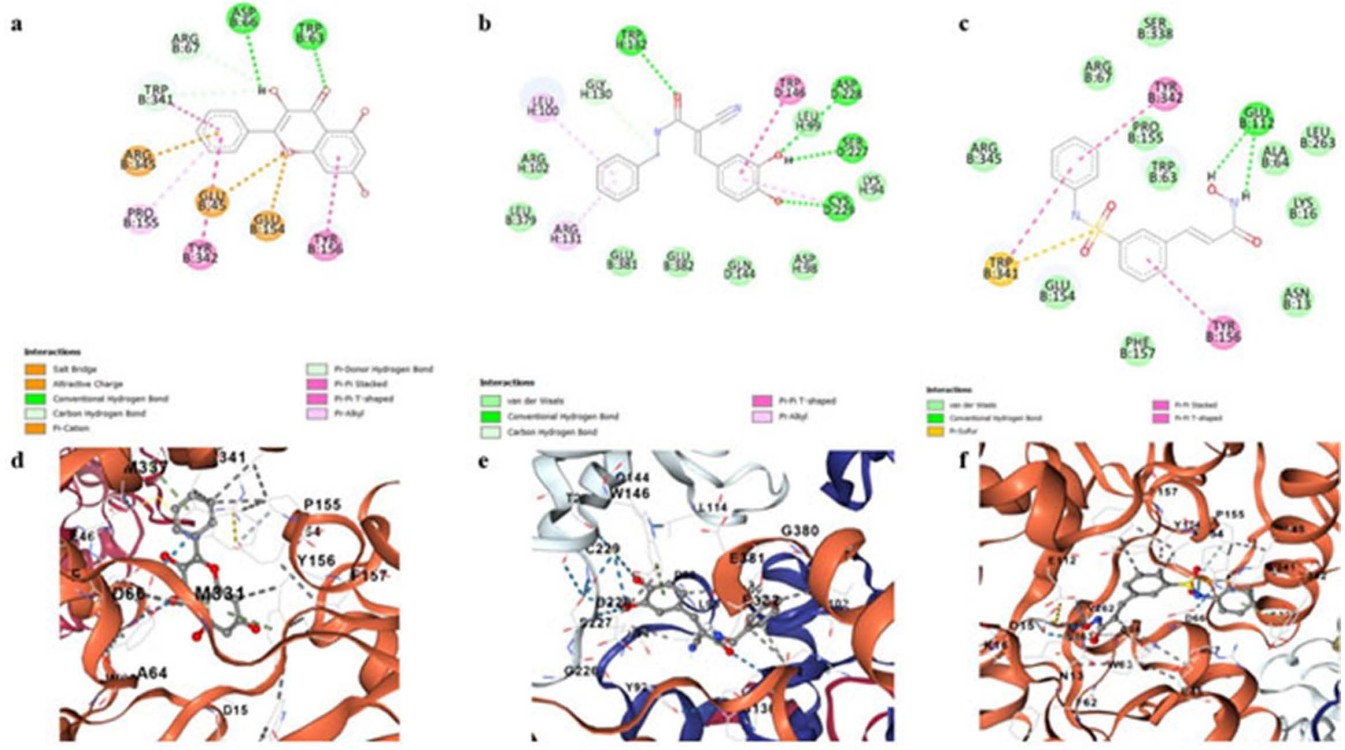

**Fig 8. Two and three dimensional binding interactions of selected compounds with the E6 protein. (a)** 2D ligand interaction of CID 5281616 with E6, **(b)** 2D ligand interaction of CID 5328779 with E6, **(c)** 2D ligand interaction of CID 6918638 with E6, **(d)** 3D ligand interaction of CID 5281616 with E6, **(e)** 3D ligand interaction of CID 5328779 with E6 and **(f)** 3D ligand interaction of CID 6918638 with E6.

CID 135540424 (Neoechinulin) formed H-bonding with GLN44, GLU46, ARG66 and ASN 53. It formed Pi- anionic interactions with ASP39. It showed attractive charge with ASP36. It displayed Pi-alkyl interactions with HIS2, TYR25 and CYS68. It showed alkyl interactions with HIS 51 (Fig 9a). CID 4788 (Phloretin) formed H-bonding with ASN53, ARG66 and CYS68. It formed carbon hydrogen bonding with HIS2 and GLY40. Pi-Pi T-shaped interactions were displayed with HIS1 and Pi stacking was shown with ASP39 (Fig 9b). CID 5280863 (Kaempferol) formed H-bonding with GLN27 and Pi stacking with HIS51. Pi-alkyl interactions were shown with CYS68. Pi-cationic interactions were shown with ARG66 and GLU46 (Fig 9c). 2D and 3D interactions of top three candidates (CID 135540424, CID 4788 and CID 5280863) with E7 are shown in Fig 9.

## Molecular dynamics

Top scorer of both E6 (Galangin) and E7 (Neoechinulin) proteins were selected for simulation studies. The root mean square deviation (RMSD), root mean square fluctuation (RMSF), and protein–ligand interactions were computed from the MD trajectory examination.

RMSD is the average change in an atom's displacement for a given frame relative to a reference frame. Figs 10 and 11 displays the RMSD of E6 and E7 Cα backbone respectively. According to the RMSD plot of E6, Galangin was tightly bonded inside E6's cavity because the RMSD grew up until 18 ns, at which point it slightly varied. The simulation showed

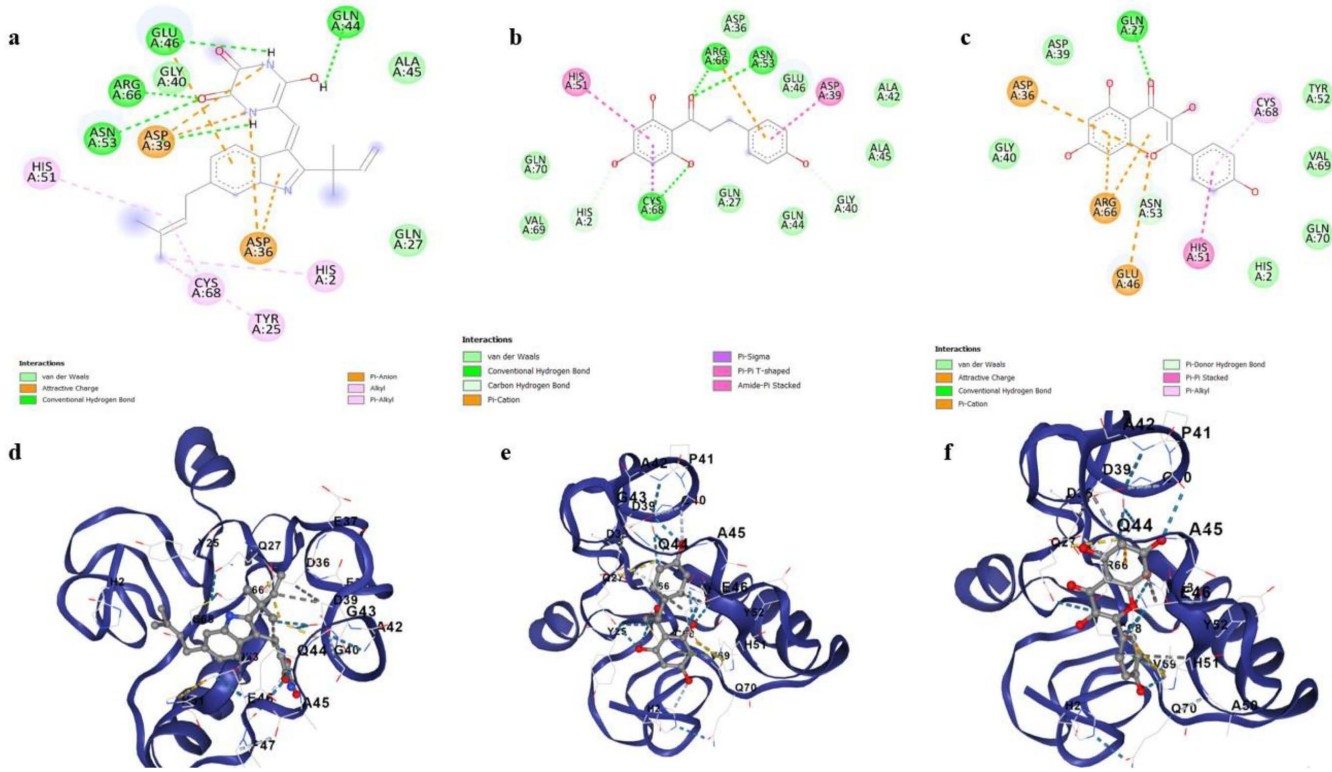

**Fig 9. Two and three dimensional binding interactions of selected compounds with the E7 protein.** (a) 2D ligand interaction of CID 135540424 with E7, **(b)** 2D ligand interaction of CID 4788 with E7, **(c)** 2D ligand interaction of CID 5280863 with E7, **(d)** 3D ligand interaction of CID 135540424 with E7, **(e)** 3D ligand interaction of CID 4788 with E7 and **(f)** 3D ligand interaction of CID 5280863 with E7.

that the Galangin–E6 complex wasn't altered, as indicated by the maximum RMSD of 2.5Å observed in the Cα backbone of E6.

RMSF is a helpful for describing local alterations in the protein chain. Figs 10 and 11 displays the RMSF plots of E6 and E7 respectively. Amino acid that showed interaction with the ligand are shown in the form of green lines. High RMSF value amino acid residues are more flexible than low RMSF value residues. The residue's flexibility is demonstrated by the maximum RMSF value of E6, which is 2.5 for amino acid 175 in the plot, while the highest RMSF value of E7 is 7.3 for amino acids 255 and 355.

Protein-ligand contact histogram of Galangin showed that it interacted with 20 amino acid residues including ASP15, LYS16, ALA64, ASP66, ARG67, GLU112, ASN 151, GLU154, TYR211, GLU215, TRP231, PHE259, VAL262, MET331, TRP341 and ARG345 as shown in Fig 10. It formed strong H-bonding with GLU112, water bridges with GLU154 and hydrophobic interactions with TYR156. Protein-ligand contact histogram of Neoechinulin showed that it interacted with 21 amino acid residues including TYR25, GLU26, GLN27, LEU28, ASN29, GLU34, GLU35, ASP36, GLU37, PRO41, GLY43, GLN44, ALA45, GLU46, PRO47, ASP48, ARG49, HIS51, ASN53, ARG 66 and CYS68 as shown in Fig 11. It formed strong hydrogen bonding with GLU35 and ARG49. It showed strong water bridges with ASP36 and ionic bonding with ARG 66.

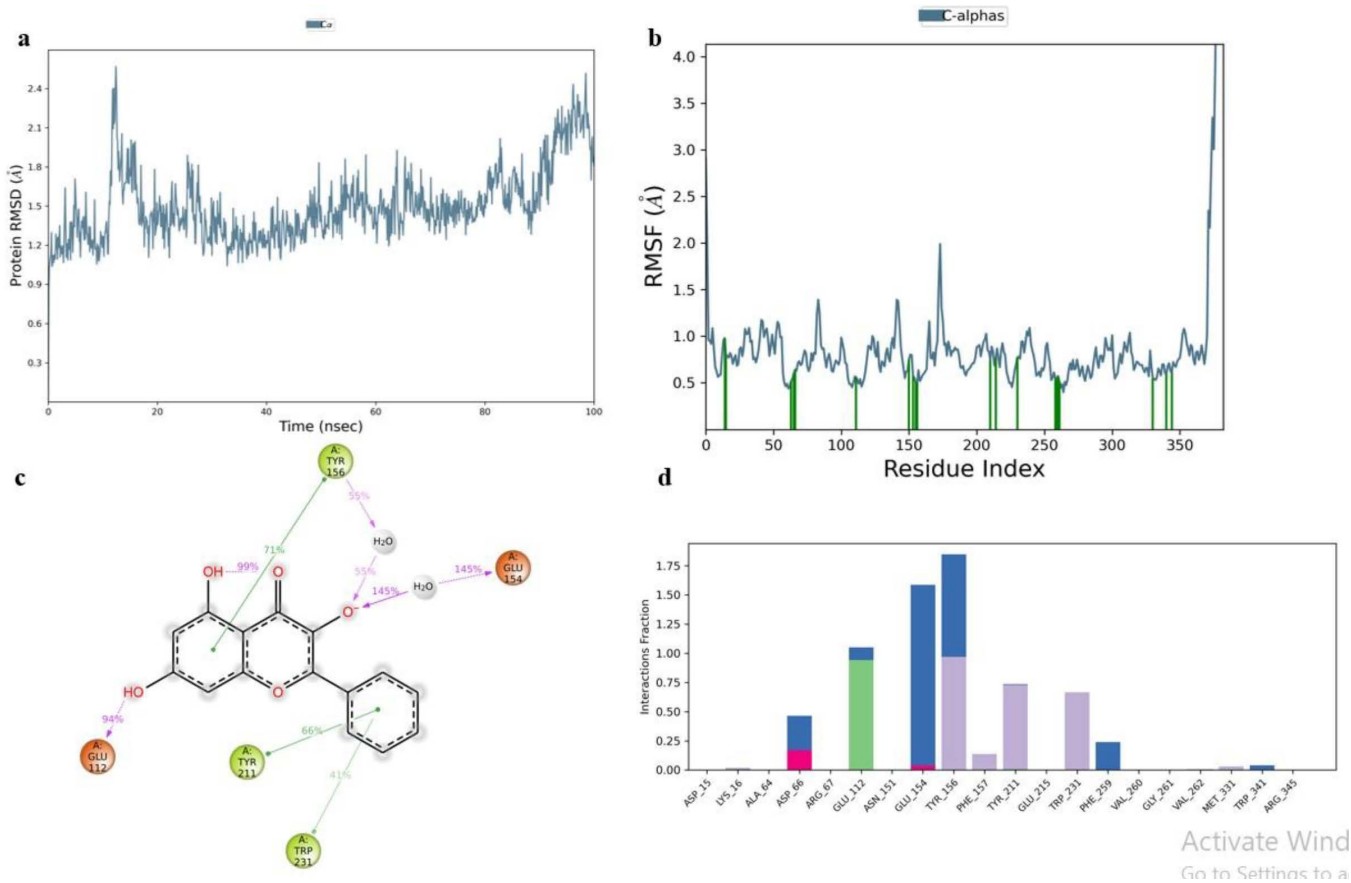

**Fig 10. Simulation analysis of Galangin-E6 complex. (a)** E6 RMSD, **(b)** E6 RMSF, **(c)** 2D ligand interaction and **(d)** Protein and ligand contact histogram.

The interactions including H-bonding, ionic interactions, hydrophobic interactions and water bridges are précised in a timeline format in the Fig 12. The number of different contacts that protein makes with the ligand during trajectory is displayed in the upper plot and which residues engage with the ligand within each trajectory framework are displayed in the bottom plot. A darker orange hue denotes the several particular contacts that some residues have with the ligand.

Ligand properties including RMSD, Intramolecular hydrogen bonds, Radius of gyration (Rg), Polar surface area, molecular surface area and solvent accessible surface area are shown in Fig 13. As shown in Fig 13, Galangin's RMSD did not fluctuate during the simulation. The Galangin–E6 complex remained unchanged during the simulation. Galangin showed an average RMSD value of 0.50 in association with E6. Neoechinulin was presumably securely bound within E7's binding pocket because the RMSD plot of E7 shows that the RMSD grew until 23 ns, at which point it little changed. Because there was no discernible dramatic variation in the plot, the Neoechinulin–E7 complex remained steady. As can be seen in Fig 13, Neoechinulin's RMSD was constant throughout the simulation. Neoechinulin's average RMSD remained 1.5 throughout. The information on the compacted nature of protein is provided by the radius of gyration (Rg) in the simulation trajectory; a larger Rg value indicates less stiffness in the system. Rg of both ligands Galangin and Neoechinulin remained stable throughout the simulation. A molecule's surface area that a water molecule can penetrate is represented by solvent accessible surface area (SASA). SASA remains consistent throughout. Slight rise in SASA indicates protein value expansion.

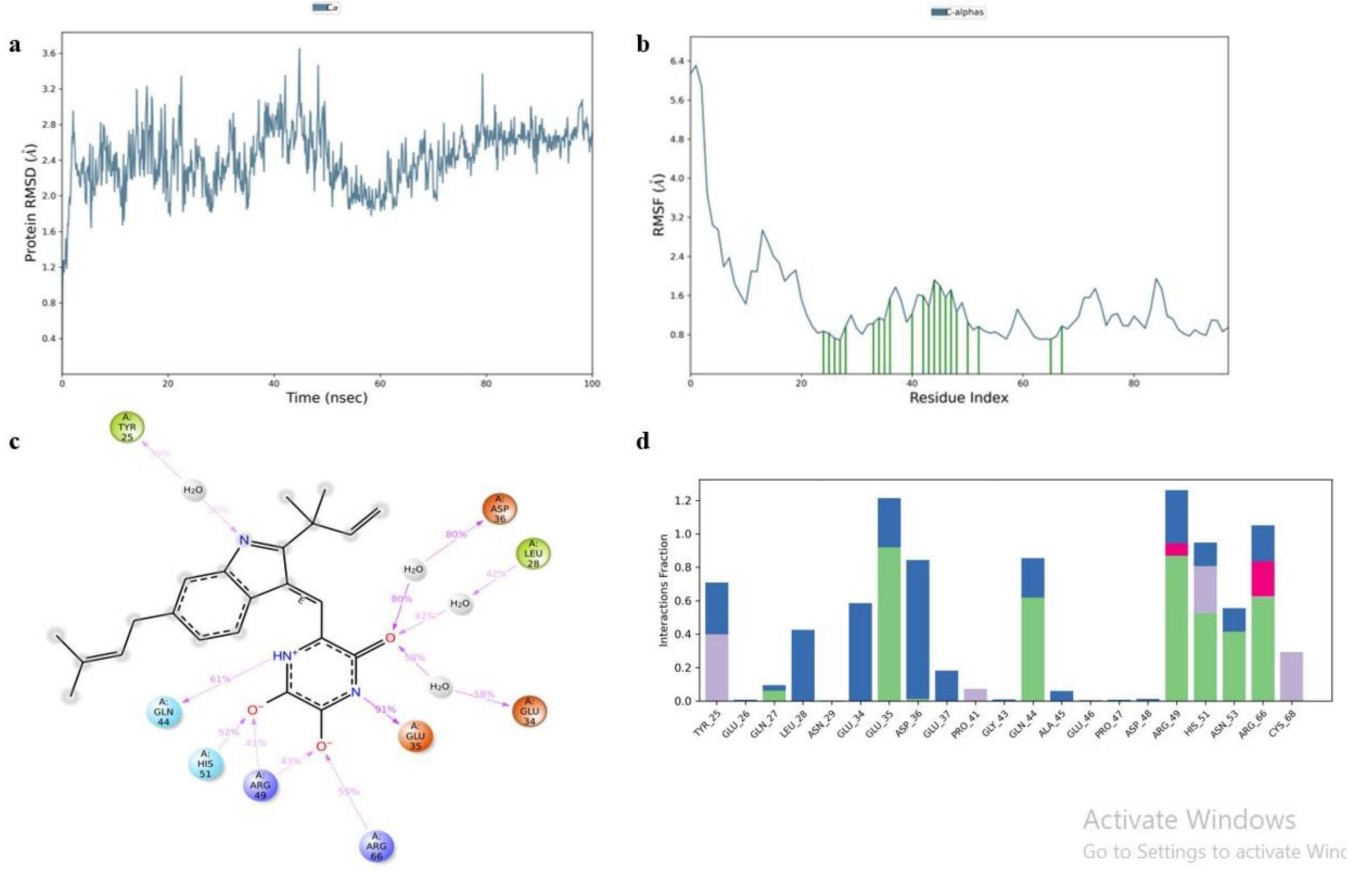

**Fig 11. Simulation analysis of Neoechinulin-E7 complex. (a)** E7 RMSD, **(b)** E7 RMSF, **(c)** 2D ligand interaction and **(d)** Protein and ligand contact histogram.

## Conclusion

This study aimed to identify potential inhibitors targeting the E6 and E7 oncogenes of Human Papilloma Virus type 16 (HPV-16) using an in silico approach. Through a rigorous virtual screening of 1000 compounds followed by a detailed molecular docking and simulation studies we identified Galangin and Neoechinulin as promising candidates. Galangin demonstrated strong binding affinity and stability towards the E6 oncogene suggesting its potential to interfere with oncogenic activity of E6. Similarly, Neoechinulin exhibited a high binding affinity and stable interactions with the E7 oncogene indicating its capacity to inhibit E7 mediated oncogenesis. These findings provide a solid foundation for further experimental validation and development of Galangin and Neoechinulin as therapeutic agents against HPV-16 related cervical cancer. Future studies should concentrate on enhancing delivery methods, gaining structural knowledge of target proteins, and experimental validation of E6 and E7 inhibitors. Successful pharmaceutical development also depends on developing combination medications, finding predictive biomarkers, and starting clinical studies. The identification of these inhibitors opens new avenues for targeted therapies against HPV-16, potentially contributing to more effective treatments and better clinical

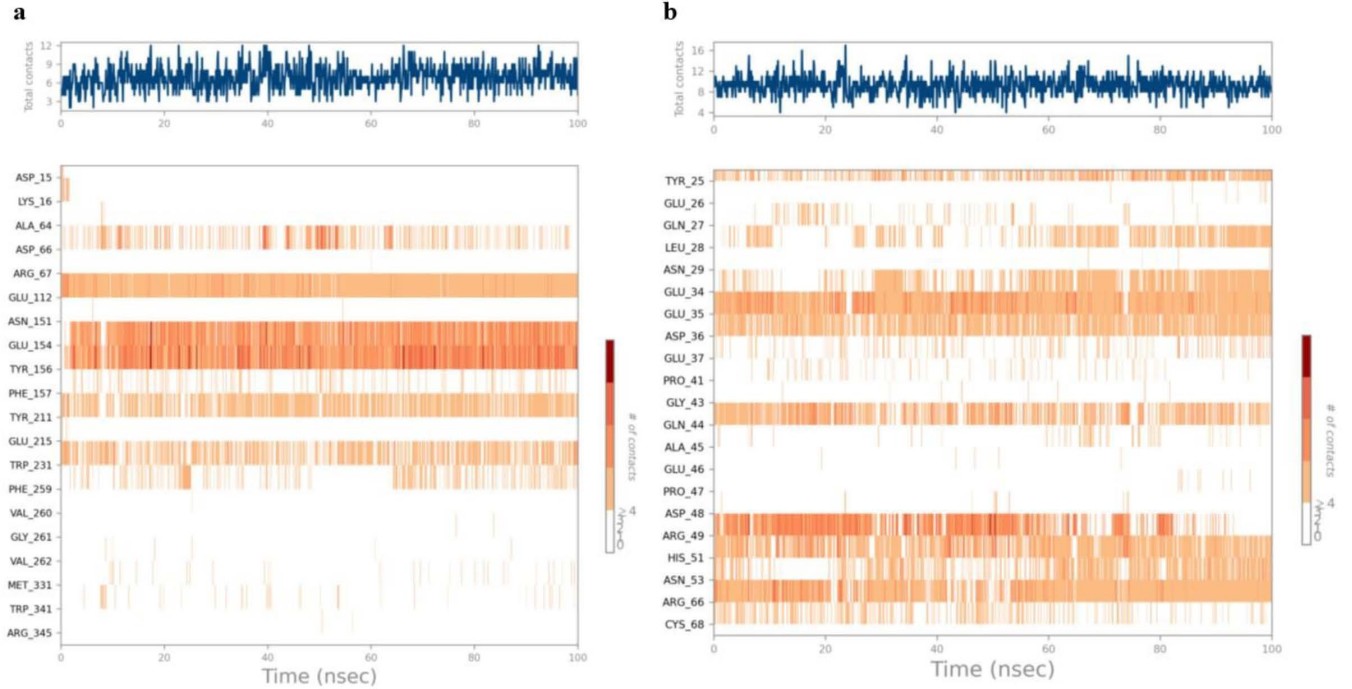

**Fig 12. Protein-ligand contacts. (a)** E6-Galangin contacts and **(b)** E7-Neoechinulin contacts.

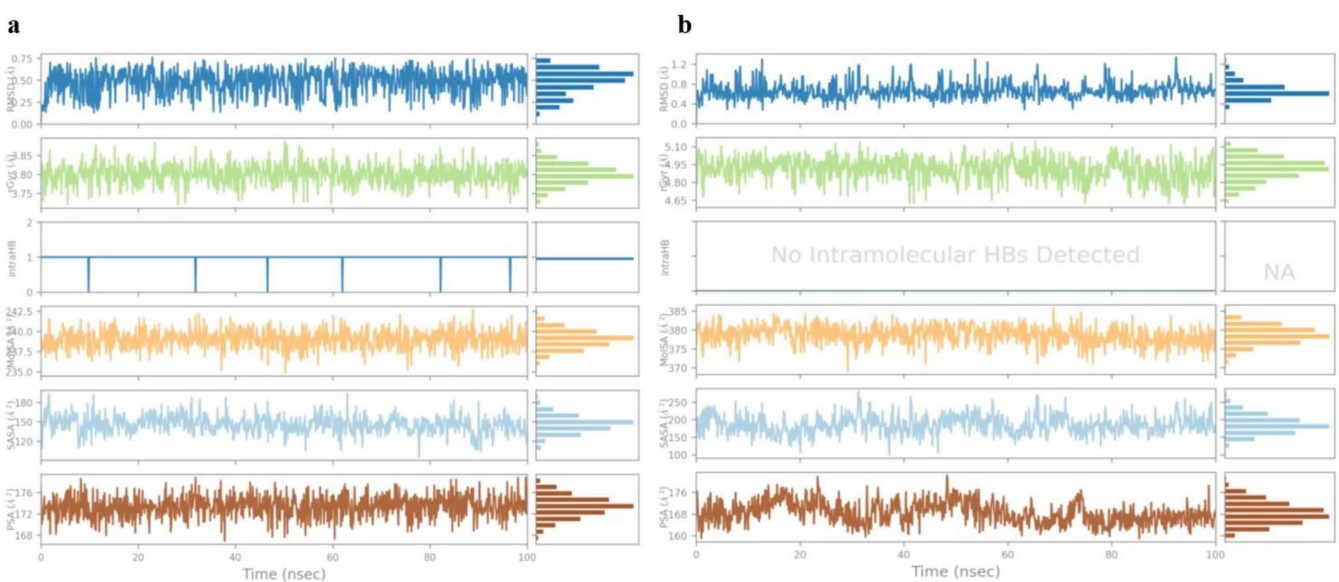

**Fig 13. Ligand properties. (a)** Galangin and **(b)** Neoechinulin.

## Supporting information

**S1 File.**
(ZIP)

## Author contributions

**Conceptualization:** Saima Younas.

**Data curation:** Saima Younas, Muhammad Umer Khan, Sadia Manzoor, Hafiz Muhammad Hammad.

**Formal analysis:** Saima Younas, Muhammad Umer Khan, Sadia Manzoor, Hafiz Muhammad Hammad, Hafiz Muzzammel Rehman.

**Investigation:** Zaryab Ikram Malik, Muhammad Umer Khan, Sadia Manzoor, Hafiz Muzzammel Rehman.

**Methodology:** Saima Younas, Zaryab Ikram Malik, Muhammad Umer Khan, Hafiz Muhammad Hammad, Hafiz Muzzammel Rehman.

**Project administration:** Saima Younas.

**Resources:** Muhammad Umer Khan, Sadia Manzoor, Hafiz Muzzammel Rehman.

**Software:** Muhammad Umer Khan, Sadia Manzoor, Hafiz Muhammad Hammad, Hafiz Muzzammel Rehman.

**Supervision:** Hafiz Muhammad Hammad, Shahina Akter.

**Validation:** Saima Younas, Shahina Akter.

**Visualization:** Saima Younas, Zaryab Ikram Malik, Hafiz Muhammad Hammad, Shahina Akter.

**Writing – original draft:** Saima Younas, Zaryab Ikram Malik.

**Writing – review & editing:** Saima Younas, Hafiz Muhammad Hammad, Shahina Akter.

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
