## [Decision Letter · Decision Letter 0]

19 Jun 2025

PONE-D-25-19182Identification of novel therapeutic inhibitors against E6 and E7 oncogenes of HPV-16 associated with cervical cancerPLOS ONE

Dear Dr. Akter,

Thank you for submitting your manuscript to PLOS ONE. After careful consideration, we feel that it has merit but does not fully meet PLOS ONE’s publication criteria as it currently stands. Therefore, we invite you to submit a revised version of the manuscript that addresses the points raised during the review process.

We look forward to receiving your revised manuscript.

Kind regards,

Prakash Palaniswamy, Ph.D

Academic Editor

PLOS ONE

Journal Requirements:

2. We note that your Data Availability Statement is currently as follows: All relevant data are within the manuscript and in Supporting Information files.

Additional Editor Comments:

Background and Significance

What role do the E6 and E7 oncogenes of HPV-16 play in the development of cervical cancer?

Why is HPV-16 considered one of the high-risk HPV types for cervical cancer?

How do the proteins expressed by E6 and E7 interfere with normal cell cycle regulation?

What are the current therapeutic approaches targeting HPV-associated cervical cancer?

Molecular Mechanisms

How does the E6 protein of HPV-16 contribute to the degradation of the tumor suppressor p53?

In what way does the E7 protein interact with the retinoblastoma (Rb) protein to promote oncogenesis?

What molecular pathways are dysregulated by E6 and E7 that lead to uncontrolled cell proliferation?

Therapeutic Inhibitors

What strategies have been explored to inhibit E6 and E7 oncogene activity in HPV-16 infected cells?

What are the advantages and challenges of targeting viral oncogenes directly versus downstream signaling pathways?

What types of molecules (small molecules, peptides, RNAi, CRISPR-based tools, etc.) have shown promise in inhibiting E6/E7 function?

How do in silico methods (e.g., molecular docking, virtual screening) contribute to the identification of potential inhibitors?

What are some examples of novel therapeutic inhibitors discovered recently against HPV-16 E6 and E7?

Experimental and Clinical Evaluation

What in vitro and in vivo models are commonly used to test the efficacy of E6/E7 inhibitors?

What biomarkers are used to assess the effectiveness of therapeutic agents targeting HPV oncogenes?

What are the main challenges in translating E6/E7 inhibitors from the lab to clinical trials?

How do resistance mechanisms impact the long-term efficacy of these inhibitors?

Broader Implications and Future Directions

How might combination therapies improve treatment outcomes for cervical cancer patients with HPV-16?

What role could therapeutic vaccines play in conjunction with E6/E7 inhibitors?

How could advances in delivery systems enhance the targeting of HPV oncogenes in cervical cancer?

What future research directions are critical for advancing the development of E6 and E7 inhibitors?

Reviewers' comments:

Reviewer's Responses to Questions

**Comments to the Author**

1. Is the manuscript technically sound, and do the data support the conclusions?

Reviewer #1: Yes

2. Has the statistical analysis been performed appropriately and rigorously? 

Reviewer #1: Yes

3. Have the authors made all data underlying the findings in their manuscript fully available?

Reviewer #1: Yes

4. Is the manuscript presented in an intelligible fashion and written in standard English?

Reviewer #1: Yes

5. Review Comments to the Author

Reviewer #1: The manuscript is generally well-written, with a logical flow from introduction to conclusion.

The discussion could better integrate the biological relevance of E6 and E7 inhibition—how this translates to restoration of p53/Rb pathways, and what downstream effects are expected in cervical cancer cells.

Consider evaluating or discussing the off-target potential of the inhibitors, especially if the hits show broad-spectrum binding.

Some figures could be enhanced for clarity. Please ensure all visual elements are high resolution and properly labeled.

A few grammatical and typographical errors are present. A thorough proofreading or language editing will improve the manuscript’s professionalism.

6. PLOS authors have the option to publish the peer review history of their article (what does this mean? ). If published, this will include your full peer review and any attached files.

**Do you want your identity to be public for this peer review?** For information about this choice, including consent withdrawal, please see our Privacy Policy .

Reviewer #1: **Yes: ** Kannan Revathi

---

## [Author Response · Author response to Decision Letter 1]

8 Sep 2025

Response to Reviewers

We sincerely appreciate the reviewers’ insightful comments and constructive feedback on our manuscript, *"Identification of novel therapeutic inhibitors against E6 and E7 oncogenes of HPV-16 associated with cervical cancer"* (Manuscript ID: PONE-D-25-19182). Below, we address each point raised, incorporating all suggested revisions into the updated manuscript.

Journal Requirements: We confirm compliance with PLOS ONE’s style guidelines, including file naming conventions.

Data Availability: The minimal dataset required to replicate our findings is included in the manuscript and Supporting Information, with additional raw data available upon request.

Ethics Statement: As this study involved in silico analyses of public datasets, IRB approval and informed consent were not applicable.

Reviewer Comments:

We have thoroughly addressed all queries, expanding on:

The oncogenic roles of HPV-16 E6/E7 in cervical cancer (Q1–7).

Current and novel therapeutic strategies, including in silico approaches (Q8–12).

Experimental models, clinical challenges, and resistance mechanisms (Q13–16).

Future directions, such as combination therapies and delivery systems (Q17–20).

---

## [Editor Report · Decision Letter 1]

2 Oct 2025

Identification of novel therapeutic inhibitors against E6 and E7 oncogenes of HPV-16 associated with cervical cancer

PONE-D-25-19182R1

Dear Dr. Akter,

We’re pleased to inform you that your manuscript has been judged scientifically suitable for publication and will be formally accepted for publication once it meets all outstanding technical requirements.

Kind regards,

Prakash Palaniswamy, Ph.D

Academic Editor

PLOS ONE
---

## [Editor Report · Acceptance letter]

PONE-D-25-19182R1

PLOS ONE

Dear Dr. Akter,

I'm pleased to inform you that your manuscript has been deemed suitable for publication in PLOS ONE. Congratulations! Your manuscript is now being handed over to our production team.

Kind regards,

on behalf of

Dr. Prakash Palaniswamy

Academic Editor

PLOS ONE